# Large-Scale Learning with Fourier Features and Tensor Decompositions

**Frederiek Wesel**
Delft Center for Systems and Control
Delft University of Technology
`f.wesel@tudelft.nl`

**Kim Batselier**
Delft Center for Systems and Control
Delft University of Technology
`k.batselier@tudelft.nl`

## Abstract

Random Fourier features provide a way to tackle large-scale machine learning problems with kernel methods. Their slow Monte Carlo convergence rate has motivated the research of *deterministic* Fourier features whose approximation error can decrease exponentially in the number of basis functions. However, due to their tensor product extension to multiple dimensions, these methods suffer heavily from the curse of dimensionality, limiting their applicability to one, two or three-dimensional scenarios. In our approach we overcome said curse of dimensionality by exploiting the tensor product structure of deterministic Fourier features, which enables us to represent the model parameters as a low-rank tensor decomposition. We derive a monotonically converging block coordinate descent algorithm with linear complexity in both the sample size and the dimensionality of the inputs for a regularized squared loss function, allowing to learn a parsimonious model in decomposed form using deterministic Fourier features. We demonstrate by means of numerical experiments how our low-rank tensor approach obtains the same performance of the corresponding nonparametric model, consistently outperforming random Fourier features.

## 1 Introduction

Kernel methods such as Support Vector Machines and Gaussian Processes are commonly used to tackle problems such as classification, regression and dimensionality reduction. Since they can be universal function approximators [13], kernel methods have received renewed attention in the last few years and have shown equivalent or superior performance to Neural Networks [22, 26, 11]. The main idea behind kernel methods is to lift the data into a higher-dimensional (or even infinite-dimensional) Reproducing Kernel Hilbert Space $\mathcal{H}$ by means of a *feature map* $\boldsymbol{\phi}\left(\cdot\right) : \mathcal{X} \rightarrow \mathcal{H}$. Considering then the pairwise similarities between the mapped data allows to tackle problems which are highly nonlinear in the original sample space $\mathcal{X}$, but linear in $\mathcal{H}$. This can be done equivalently by considering a *kernel function* $k\left(\cdot,\cdot\right) : \mathcal{X} \times \mathcal{X} \rightarrow \mathbb{R}$ such that $\langle \boldsymbol{\phi}\left(\boldsymbol{x}\right), \boldsymbol{\phi}\left(\boldsymbol{x'}\right)\rangle = k\left(\boldsymbol{x},\boldsymbol{x'}\right)$ and performing thus said mapping implicitly. Although effective at learning nonlinear patterns in the data, kernel methods are known to scale poorly as the number of data points $N$ increases. For example, when considering Kernel Ridge Regression (KRR), Gaussian Process Regression (GPR) [33] or Least-Squares Support Vector Machine (LS-SVM) [41, 42] training usually consist in inverting the *Gram matrix* $k_{ij} = k\left(\boldsymbol{x}_i,\boldsymbol{x}_j\right)$, which encodes the pairwise relation between all data. As a consequence, the associated storage complexity is $\mathcal{O}\left(N^2\right)$ and the computational complexity is $\mathcal{O}\left(N^3\right)$, rendering these methods unfeasible for large data. In order to lower the computational cost, *data dependent* methods approximate the kernel function by means of $M$ data dependent basis functions. Due to their reduced-rank formulation, the computational complexity is reduced to $\mathcal{O}\left(NM^2\right)$ for $N \gg M$. However, e.g. for the Nyström method [46], the convergence rate is

35th Conference on Neural Information Processing Systems (NeurIPS 2021).

only of $\mathcal{O}\left(1/\sqrt{M}\right)$ [7] limiting its effectiveness. As the name suggests, *data independent* methods approximate the kernel function by $M$ data independent basis functions. A good example is the celebrated Random Fourier Features approach by Rahimi and Recht [31], where the authors propose for stationary kernels a low-dimensional *random* mapping $\boldsymbol{z}\left(\cdot\right):\mathbb{R}^D \to \mathbb{R}^M$ such that

$$k\left(\boldsymbol{x},\boldsymbol{x}'\right) = \langle \boldsymbol{\phi}\left(\boldsymbol{x}\right), \boldsymbol{\phi}\left(\boldsymbol{x}'\right)\rangle \approx \langle \boldsymbol{z}\left(\boldsymbol{x}\right), \boldsymbol{z}\left(\boldsymbol{x}'\right)\rangle. \tag{1}$$

As in the case of data dependent methods, the reduced-rank formulation allows for a computational complexity of $\mathcal{O}\left(NM^2\right)$ for $N \gg M$. Probabilistic error bounds on the approximation are provided which result in a convergence rate of $\mathcal{O}\left(1/\sqrt{M}\right)$, which is again the Monte Carlo rate.

Improvements in this sense were achieved by considering *deterministic* features resulting from dense quadrature methods [5, 25, 36] and kernel eigenfunctions [16, 39]. These methods are able to achieve *exponentially* decreasing upper bounds on uniform convergence guarantees when certain conditions are met. However, for a $D$-dimensional input space these methods take the tensor product of $D$ vectors, resulting in an exponential increase in the number of basis functions and thus of model weights, effectively limiting the applicability of deterministic features to low-dimensional data. In this work we consider deterministic features. In order to take advantage of the tensor product structure which arises when mapping the inputs to $\mathcal{H}$, we represent the weights as a low-rank tensor. This allows us to *learn* the inter-modal relations in the tensor product of (low-dimensional) Hilbert spaces, avoiding the exponential computational and storage complexities in $D$. In this way we are able to obtain a *linear* computational complexity in both the number of samples *and* in the input dimension during training, without having to resort to the use of sparse grids or additive modeling of kernels.

The main contribution of this work is in lifting the curse of dimensionality affecting deterministic Fourier features by modeling the weights as a low-rank tensor. This enables the efficient solution of large-scale and high-dimensional kernel learning problems. We derive an iterative algorithm under the exemplifying case of regularized squared loss and test it on regression and classification problems.

## 2 Related work

Fourier Features (FF) are a collection of data independent methods that leverage Bochner's theorem [35] from harmonic analysis to approximate stationary kernels by numerical integration of their spectral density $p\left(\cdot\right)$:

$$k\left(\boldsymbol{x},\boldsymbol{x}'\right) \overset{\text{Stationarity}}{:=} k\left(\boldsymbol{x}-\boldsymbol{x}'\right) \overset{\text{Bochner}}{=} \int p\left(\boldsymbol{\omega}\right)\exp\left(\langle i\boldsymbol{\omega},\left(\boldsymbol{x}-\boldsymbol{x}'\right)\rangle\right)d\boldsymbol{\omega} \overset{\text{FF}}{\approx} \langle z\left(\boldsymbol{x}\right), z\left(\boldsymbol{x}'\right)\rangle. \tag{2}$$

Rahimi and Recht [31] proposed to approximate the integral by Monte Carlo integration i.e. by drawing $M$ random frequencies $\boldsymbol{\omega} \sim p\left(\cdot\right)$. In their work they show how the method converges uniformly at the Monte Carlo rate. See [23] for an overview of random FF. In order to achieve faster convergence and a lower sample complexity, a multitude of approaches that rely on deterministic numerical quadrature of the Fourier integral were developed. These methods generally consider *product kernels* whose spectral density factors in the frequency domain, which enables in turn to factor the Fourier integral. The resulting deterministic Fourier features are then the tensor product of $D$ one-dimensional deterministic features.

For example, in [5] the authors give an analysis of the sample complexity of features resulting from dense *Gaussian Quadrature* (GQ). In [25] the authors present a similar quadrature approach for kernels whose spectral density factors over the dimensions of the inputs. They provide an explicit construction for their *Quadrature Fourier Features* relying on a dense Cartesian grid, and note that their method, as well as GQ, can attain exponentially decreasing uniform convergence bounds in the total number of basis functions per dimensions $\hat{M}$ [25, Theorem 1]. To avoid the curse of dimensionality, they make use of additive modeling of kernels. *Variational Fourier Features* (VFF) [16] are derived probabilistically in a one-dimensional setting for Matérn kernels by projecting a Gaussian Process onto a set of Fourier basis. An extension to multiple dimensions when considering product kernels is then achieved by taking the tensor product of the one-dimensional features, incurring however again in exponentially rising computational costs in $D$. In what they call *Hilbert-GP* [39] the authors diagonalize stationary kernels in terms of the eigenvalues and eigenfunctions of the Laplace operator with Dirichlet boundary conditions. Due to the multiplicative structure of the

eigenfunctions of the Laplace operator, the complexity of the basis function increases exponentially in $D$ when one considers product kernels. Like with other deterministic approximations, bounds on the uniform convergence error which decrease exponentially in $\hat{M}$ can be achieved [39, Theorem 8].

Tensor decompositions have been used extensively in machine learning in order to benefit from structure in data [38], in particular to obtain an efficient representation of the model parameters. In the context of Gaussian Process Regression, the tensor product structure which arises when the inputs are located on a Cartesian grid have been exploited to speedup inference [12]. In their variational inference framework, [19] proposed to parameterize the posterior mean of a Gaussian Process by means of a tensor decomposition in order to exploit the tensor product structure which arises when interpolating the kernel function. In the context of neural networks, [27] proposed to represent the weights in deep neural networks as a tensor decomposition to speed up training. A similar approach was carried out in case of recurrent neural networks [48] [43].

Tensor decompositions have also been used to learn from simple features. In [45] multivariate functions are learned from a Fourier basis, which corresponds to assuming a uniform spectral density in (2). Motivated by spin vectors in quantum mechanics, [40] consider trigonometric feature maps which they argue induce uninformative kernels. On a similar note, [28] and [1] leverage the tensor product structure of polynomial mappings to induce a polynomial kernel and consider model parameters in decomposed form. While these existing approaches are able to overcome the curse of dimensionality affecting tensor product feature maps by modeling the parameter tensor as a tensor network, they consider simple and empirical feature maps which induce uninformative kernels. In the following section we show how deterministic Fourier features can be used for supervised learning in both large and high-dimensional scenarios by assuming that the model weights are a low-rank tensor, thereby linking tensor decompositions with stationary product kernels.

## 3 Learning with deterministic Fourier features

### 3.1 Notation

All tensor operations and mappings can be formulated over both the complex and the real field. For the remainder of this article real-valued mappings will be considered. Vectors and matrices are denoted by boldface lowercase and uppercase letters, respectively. Higher-order tensors are denoted by boldface calligraphic letters. Tensor entries are always written as lightface lowercase letters with a subscript index notation. For example, element $i_1, i_2, i_3$ of the third-order tensor $\boldsymbol{\mathcal{A}} \in \mathbb{R}^{I_1 \times I_2 \times I_3}$ is written as $a_{i_1 i_2 i_3}$. The vectorization $\text{vec}(\boldsymbol{\mathcal{A}})$ of a tensor $\boldsymbol{\mathcal{A}} \in \mathbb{R}^{I_1 \times I_2 \times \cdots \times I_D}$ is the vector such that

$$\text{vec}(\boldsymbol{\mathcal{A}})_i = a_{i_1 i_2 \cdots i_D},$$

where the relationship between the linear index $i$ and $i_1 i_2 \ldots i_D$ is given by

$$i = i_1 + \sum_{d=2}^{D} (i_d - 1) \prod_{k=1}^{d-1} I_k.$$

The Frobenius inner product between tensors $\boldsymbol{\mathcal{A}} \in \mathbb{R}^{I_1 \times I_2 \times \cdots \times I_D}$ and $\boldsymbol{\mathcal{B}} \in \mathbb{R}^{I_1 \times I_2 \times \cdots \times I_D}$ is defined as

$$\langle \boldsymbol{\mathcal{A}}, \boldsymbol{\mathcal{B}} \rangle_{\text{F}} := \sum_{i_1=1}^{I_1} \sum_{i_2=1}^{I_2} \cdots \sum_{i_D=1}^{I_D} a_{i_1 i_2 \cdots i_D} b_{i_1 i_2 \cdots i_D} = \text{vec}(\boldsymbol{\mathcal{A}})^{\text{T}} \text{vec}(\boldsymbol{\mathcal{B}}).$$

Depending on the context, the symbol $\otimes$ either denotes the tensor outer product or the matrix Kronecker product. The Khatri-Rao product $\boldsymbol{A} \circledast \boldsymbol{B}$ of the matrices $\boldsymbol{A} \in \mathbb{R}^{N \times R}$ and $\boldsymbol{B} \in \mathbb{R}^{M \times R}$ is the $NM \times R$ matrix that is obtained by taking the column-wise Kronecker product. The Hadamard (element-wise) matrix product of $\boldsymbol{A} \in \mathbb{R}^{N \times R}$ and $\boldsymbol{B} \in \mathbb{R}^{N \times R}$ is denoted by $\boldsymbol{A} \odot \boldsymbol{B}$.

### 3.2 The model

In this article we assume models of the form

$$f(\boldsymbol{x}) = \langle \boldsymbol{w}, \boldsymbol{\phi}(\boldsymbol{x}) \rangle, \quad y = f(\boldsymbol{x}) + \epsilon. \tag{3}$$

Here $\boldsymbol{x} \in \mathbb{R}^D$ is the input vector, $\boldsymbol{\phi}\left(\cdot\right): \mathbb{R}^D \to \mathcal{H}$ is a feature map, $\boldsymbol{w} \in \mathcal{H}$ are the model weights, $y \in \mathbb{R}$ is the corresponding available observation corrupted by a zero mean i.i.d. noise $\epsilon$. Learning such a kernelized model consists in finding as set of weights $\boldsymbol{w}$ such that:

$$\sum_{n=1}^{N} L\left(y_n, f\left(\boldsymbol{x}_n\right)\right) + \lambda \left\langle \boldsymbol{w}, \boldsymbol{w} \right\rangle_{\mathrm{p}}, \tag{4}$$

is minimized. Here, $L\left(\cdot, \cdot\right): \mathbb{R} \times \mathbb{R} \to \mathbb{R}_+$ is a symmetric and positive loss function and $\lambda \left\langle \boldsymbol{w}, \boldsymbol{w} \right\rangle_{\mathrm{p}}$ is a $p$-th norm regularization term. A variety of *primal* machine learning formulations arise when considering different combinations of loss functions and regularization terms. For example, considering $p = 2$ and squared loss leads to Kernel Ridge Regression [42], while considering hinge loss leads to SVM [4], and so on. Furthermore, applying the *kernel trick* when considering $\boldsymbol{\phi}\left(\cdot\right)$ recovers the nonparametric formulation which, depending on the choice of kernel, enables to perform inference using infinitely many basis functions with a computational complexity of $\mathcal{O}(N^3)$. Considering instead a low-dimensional finite mapping $\boldsymbol{z}\left(\cdot\right): \mathcal{X} \to \mathbb{R}^M$ such as Random Fourier Features or the Nyström method leads a primal approach and to computational savings with a computational complexity of $\mathcal{O}\left(NM^2\right)$ for $N \gg M$. However, as already discussed, these low-dimensional random mappings converge at the slow Monte Carlo rate, motivating our approach.

In order to approximate the kernel function with faster and possibly *exponential* convergence, we consider deterministic, finite-dimensional features which are the tensor product of $D$ vectors. For a given $D$-dimensional input point $\boldsymbol{x}$ we define the deterministic feature mapping $\boldsymbol{\mathcal{Z}}\left(\cdot\right): \mathbb{R}^D \to \mathbb{R}^{\hat{M}_1} \otimes \mathbb{R}^{\hat{M}_2} \otimes \cdots \otimes \mathbb{R}^{\hat{M}_D}$ as

$$\boldsymbol{\mathcal{Z}}\left(\boldsymbol{x}\right) = \boldsymbol{z}^{(1)}\left(x_1\right) \otimes \boldsymbol{z}^{(2)}\left(x_2\right) \cdots \otimes \boldsymbol{z}^{(D)}\left(x_D\right), \tag{5}$$

where $\boldsymbol{z}^{(d)}$ is a deterministic mapping applied to the $d$-th dimension. The dimension of the feature space is denoted $M = \prod_{d=1}^{D} \hat{M}_d$. Without loss of generality it is from here on assumed that $\hat{M}_1 = \cdots = \hat{M}_D = \hat{M}$ such that $M = \hat{M}^D$. It is easy to verify by applying the kernel trick that the features of (5) yield product kernels. The mapping $\boldsymbol{\mathcal{Z}}(\cdot)$ encompasses for instance the mappings derived by quadrature [5, 25, 36] and by projection [16, 39]. Note that these mappings cover many popular kernels such as the Gaussian and Matérn kernels. To give a concrete example, in the framework of A. Solin and S. Särkkä [39, Equation 60], for input data centered in a hyperbox $\mathcal{X} = [-U_1, U_1] \otimes \cdots \otimes [-U_D, U_D]$, the Gaussian kernel is approximated by means of $D$ tensor products of $\hat{M}$ weighted sinusoidal basis functions with frequencies lying on a harmonic scale such that:

$$\left(\boldsymbol{z}^{(d)}\left(x_d\right)\right)_{i_d} = \frac{1}{\sqrt{U_d}} \, p\left(\frac{\pi i_d}{2U_d}\right) \, \sin\left(\frac{\pi i_d\left(x_d + U_d\right)}{2U_d}\right), \ i_d = 1, \ldots, \hat{M}. \tag{6}$$

Here $p\left(\cdot\right)$ is the spectral density of the Gaussian kernel with one-dimensional inputs, which is known in closed-form [33, page 83]. This deterministic mapping then approximates the Gaussian kernel function [39, Equation 59] such that

$$k\left(\boldsymbol{x}, \boldsymbol{x}'\right) \approx \left\langle \mathrm{vec}\left(\boldsymbol{\mathcal{Z}}\left(\boldsymbol{x}\right)\right), \mathrm{vec}\left(\boldsymbol{\mathcal{Z}}\left(\boldsymbol{x}'\right)\right) \right\rangle = \left\langle \boldsymbol{\mathcal{Z}}\left(\boldsymbol{x}\right), \boldsymbol{\mathcal{Z}}\left(\boldsymbol{x}'\right) \right\rangle_{\mathrm{F}},$$

and converges uniformly with exponentially decreasing bounds [39, Theorem 8]. We therefore use $\boldsymbol{\mathcal{Z}}(\cdot)$ instead of $\boldsymbol{\phi}(\cdot)$ in (3) to obtain

$$f\left(\boldsymbol{x}\right) = \left\langle \boldsymbol{w}, \mathrm{vec}(\boldsymbol{\mathcal{Z}}(\boldsymbol{x})) \right\rangle = \left\langle \boldsymbol{\mathcal{W}}, \boldsymbol{\mathcal{Z}}(\boldsymbol{x}) \right\rangle_{\mathrm{F}}, \tag{7}$$

where the weight vector $\boldsymbol{w}$ has been reshaped into a $D$-dimensional tensor $\boldsymbol{\mathcal{W}} \in \mathbb{R}^{\hat{M}_1 \times \hat{M}_2 \times \cdots \times \hat{M}_D}$. Learning the exponential number of model parameters $\boldsymbol{\mathcal{W}}$ in (7) under a hinge loss leads to Support Vector Machines, while considering a squared loss leads to the primal formulation of Kernel Ridge Regression, which we will consider as exemplifying case from here on:

$$\min_{\boldsymbol{\mathcal{W}}} \sum_{n=1}^{N} \left(y_n - \left\langle \boldsymbol{\mathcal{W}}, \boldsymbol{\mathcal{Z}}\left(\boldsymbol{x}_n\right) \right\rangle_{\mathrm{F}}\right)^2 + \lambda \left\langle \boldsymbol{\mathcal{W}}, \boldsymbol{\mathcal{W}} \right\rangle_{\mathrm{F}}. \tag{8}$$

Since the number of elements $M$ in $\boldsymbol{\mathcal{W}}$ and $\boldsymbol{\mathcal{Z}}$ grows exponentially in $D$, this primal approach is advantageous compared to the nonparametric *dual* approach only if $\hat{M}^D \ll N$, limiting it to low-dimensional inputs. In order to lift this curse of dimensionality, we propose to represent and to learn $\boldsymbol{\mathcal{W}}$ directly as a low-rank tensor decomposition. A low-rank tensor decomposition allows us to exploit redundancies in $\boldsymbol{\mathcal{W}}$ in order to obtain a parsimonious model with a storage complexity that scales linearly in both $\hat{M}$ and $D$. The low-rank structure will, as explicitly shown in the experiments, act as a form of regularization by limiting the total number of degrees of freedom of the model.

### 3.3 Low-rank tensor decompositions

Tensor decompositions (also called tensor networks) can be seen as generalizations of the Singular Value Decomposition (SVD) of a matrix to tensors [20]. Three common tensor decompositions are the Tucker decomposition [6, 44], the Tensor Train decomposition [29] and the Canonical Polyadic Decomposition (CPD) [17, 21], where each of them encompasses different properties of the matrix SVD. In this subsection we briefly discuss these three decompositions and list both their advantages and disadvantages for modeling $\mathcal{W}$ in (8). For a detailed exposition on tensor decompositions we refer the reader to [2] and the references therein. An important property of tensor decompositions is that they can always be written linearly in their components, which implies that applying a block coordinate descent algorithm to solve (8) results in a series of linear least-squares problems.

A rank-$R$ CPD of $\mathcal{W}$ consists of $D$ factor matrices $\boldsymbol{W}^{(d)} \in \mathbb{R}^{\hat{M} \times R}$ and a vector $\boldsymbol{s} \in \mathbb{R}^R$ such that

$$\boldsymbol{w} = \left( \boldsymbol{W}^{(1)} \circledast \boldsymbol{W}^{(2)} \circledast \cdots \circledast \boldsymbol{W}^{(D)} \right) \boldsymbol{s} = \sum_{r=1}^{R} \boldsymbol{w}_r^{(1)} \otimes \boldsymbol{w}_r^{(2)} \otimes \cdots \otimes \boldsymbol{w}_r^{(D)}.$$

Note that the entries of the $\boldsymbol{s}$ vector were absorbed in one of the factor matrices when writing the CPD as a sum of $R$ terms. The CPD has been shown, in contrast to matrix factorizations, to be unique under mild conditions [37]. The storage complexity comes mainly from the $D$ factor matrices and is therefore $\mathcal{O}(R\hat{M}D)$. The Tucker decomposition generalizes the CPD in two ways. First, the Khatri-Rao product is replaced with a Kronecker product. The $\boldsymbol{s}$ vector has to grow in length accordingly from $R$ to $R^D$. Second, the $R$-dimension of each of the factor matrices is allowed to vary, resulting in a multi-linear rank $(R_1, R_2, \ldots, R_D)$:

$$\boldsymbol{w} = \left( \boldsymbol{W}^{(1)} \otimes \boldsymbol{W}^{(2)} \otimes \cdots \otimes \boldsymbol{W}^{(D)} \right) \boldsymbol{s}.$$

The Tucker decomposition is inherently non-unique and its storage complexity $\mathcal{O}(R^D)$ is dominated by the $\boldsymbol{s}$ vector, which limits its use to low-dimensional input data. For this reason we do not consider the Tucker decomposition any further in this article.

The Tensor Train (TT) decomposition consists of $D$ third-order tensors $\mathcal{W}^{(d)} \in \mathbb{R}^{R_d \times \hat{M} \times R_{d+1}}$ such that

$$w_{i_1 i_2 \cdots i_D} = \sum_{r_1=1}^{R_1} \cdots \sum_{r_{D+1}=1}^{R_{D+1}} w_{r_1 i_1 r_2}^{(1)} \cdots w_{r_D i_D r_{D+1}}^{(D)}. \tag{9}$$

The auxiliary dimensions $R_1, R_2, \ldots, R_{D+1}$ are called the TT-ranks. In order to ensure that the right-hand side of (9) is a scalar, the boundary condition $R_1 = R_{D+1} = 1$ is enforced. The TT decomposition is, just like the Tucker decomposition, non-unique and its storage complexity $R^2 \hat{M} D$ is due to the $D$ tensor components $\mathcal{W}^{(d)}$. Considering their storage complexity, both the CPD and TT decomposition are viable candidates to replace $\mathcal{W}$ in (8). The CP-rank $R$ and TT-ranks $R_2, \ldots, R_D$ are additional hyperparameters, which favors the CPD in practice. For the TT, one could choose $R_2 = R_3 = \cdots = R_D$ to reduce the number of additional hyperparameters but this constraint turns out in practice to lead to suboptimal results. For these reasons we limit the discussion of the learning algorithm to the CPD case.

### 3.4 Tensor learning with deterministic Fourier features

We now wish to minimize the standard regularized squared loss function as in (8) with the additional constraint that the weight tensor has a rank-$R$ CPD structure:

$$\min_{\mathcal{W}} \quad \sum_{n=1}^{N} \left( y_n - \langle \mathcal{W}, \mathcal{Z}\left(\boldsymbol{x}_n\right) \rangle_{\mathrm{F}} \right)^2 + \lambda \left\langle \mathcal{W}, \mathcal{W} \right\rangle_{\mathrm{F}}, \tag{10}$$

$$\text{subject to} \quad \text{CP-rank}\left(\mathcal{W}\right) = R. \tag{11}$$

Note that if $R$ equals the true CP-rank of the underlying weight tensor then the exact solution of (8) would be obtained. In practice a low-rank solution for $\mathcal{W}$ achieves a sufficiently complex decision boundary that is practically indistinguishable from the full-rank solution, as is demonstrated in the

experiments. Imposing the rank-$R$ reduces the number of unknowns from $\hat{M}^D$ to $R\hat{M}D$ and allows the application of a block coordinate descent algorithm (also known as alternating linear scheme), which is shown to converge monotonically [3, 18, 34]. Each of the factor matrices $\boldsymbol{W}^{(d)}$ is optimized in an iterative fashion while keeping the others fixed. Such a factor matrix update is obtained by solving a linear least-squares problem with $\hat{M}R$ unknowns. In what follows we derive the linear problem for the $d$-th factor matrix $\boldsymbol{W}^{(d)}$. The data-fitting term $\langle \boldsymbol{\mathcal{W}}, \boldsymbol{\mathcal{Z}}(\boldsymbol{x})\rangle_{\mathrm{F}}$ can be rewritten linearly in terms of the unknown factor matrix as

$$\langle \boldsymbol{\mathcal{W}}, \boldsymbol{\mathcal{Z}}(\boldsymbol{x})\rangle_{\mathrm{F}} = \left\langle \mathrm{vec}\left(\boldsymbol{W}^{(d)}\right), \boldsymbol{g}^{(d)}\left(\boldsymbol{x}\right)\right\rangle, \tag{12}$$

where $\boldsymbol{g}^{(d)}\left(\boldsymbol{x}\right) \coloneqq \boldsymbol{z}^{(d)} \otimes \left(\boldsymbol{z}^{(1)^{\mathrm{T}}}\boldsymbol{W}^{(1)^{\mathrm{T}}} \odot \cdots \odot \boldsymbol{z}^{(D)^{\mathrm{T}}}\boldsymbol{W}^{(D)^{\mathrm{T}}}\right)$. Similarly, for the regularization term:

$$\langle \boldsymbol{\mathcal{W}}, \boldsymbol{\mathcal{W}}\rangle_{\mathrm{F}} = \left\langle \mathrm{vec}\left(\boldsymbol{W}^{(d)^{\mathrm{T}}}\boldsymbol{W}^{(d)}\right), \mathrm{vec}\left(\boldsymbol{H}^{(d)}\right)\right\rangle. \tag{13}$$

Here $\boldsymbol{H}^{(d)} \coloneqq \left(\boldsymbol{W}^{(1)^{\mathrm{T}}}\boldsymbol{W}^{(1)} \odot \cdots \odot \boldsymbol{W}^{(D)^{\mathrm{T}}}\boldsymbol{W}^{(D)}\right)$. Derivations of (12) and (13) can be found in Appendix. Substitution of the expressions (12) and (13) into (10) and (11) leads to a linear least-squares problem for $\mathrm{vec}\left(\boldsymbol{W}^{(d)}\right)$:

$$\min_{\mathrm{vec}\left(\boldsymbol{W}^{(d)}\right)} \sum_{n=1}^{N} \left(y_n - \left\langle \mathrm{vec}\left(\boldsymbol{W}^{(d)}\right), \boldsymbol{g}^{(d)}\left(\boldsymbol{x}_n\right)\right\rangle\right)^2 + \lambda \left\langle \mathrm{vec}\left(\boldsymbol{W}^{(d)^{\mathrm{T}}}\boldsymbol{W}^{(d)}\right), \mathrm{vec}\left(\boldsymbol{H}^{(d)}\right)\right\rangle. \tag{14}$$

Its unique solution can be computed exactly in an efficient manner by solving the normal equations, requiring $N\hat{M}^2R^2 + \hat{M}^3R^3$ operations. Since we are solving a non-convex optimization problem that consists of a series of convex and exactly solvable sub-problems, our algorithm is indeed monotonically decreasing, and, although it is not guaranteed to converge to the global optimum, it is the standard choice when dealing with tensors [3, 20]. The total computational complexity of the algorithm when $N \gg \hat{M}R$ is then $\mathcal{O}(ND\hat{M}^2R^2)$, rendering it suitable for problems which are large in both $N$ and $D$, provided that $R$ and $\hat{M}$ are small. The necessary memory is equal to $R\hat{M}D + 2R^2\hat{M}^2 + 2R\hat{M}$ (storing the weight tensor $\boldsymbol{\mathcal{W}}$ in decomposed form, the rank-$R\hat{M}$ Gram matrix and regularization matrix, the transformed responses and the solution of the linear system), leading to a storage complexity of $\mathcal{O}\left(R^2\hat{M}^2\right)$ for $R\hat{M} \gg D$. Notably the cost is independent of $N$, which allows processing large data on modest hardware.

When learning, the selection of $\lambda$ and of the kernel-related hyperparameters can be carried out by standard methods such as cross-validation. The choice of $\hat{M}$ and the additional hyperparameter $R$ which we introduce are directly linked with the available computational budget. One should in fact choose $\hat{M}$ so that the model has access to a sufficiently complex set of basis functions to learn from. In practice we notice that for our choices of kernel function hyperparameters, at most $\hat{M} = 40$ basis functions per dimension suffice. $R$ can then be fixed accordingly in order to match the computational budget at hand. As we will show in the next section, learning is possible with small values of $\hat{M}$ and $R$.

## 4   Numerical experiments

We implemented our Tensor-Kernel Ridge Regression (T-KRR) algorithm in Mathworks MATLAB 2021a (Update 1) [24] and tested it on several regression and classification problems. Our implementation can be freely downloaded from `https://github.com/fwesel/T-KRR` and allows reproduction of all experiments in this section. In our implementation we avoid constructing $\boldsymbol{g}^{(d)}\left(\cdot\right)$ and $\boldsymbol{H}^{(d)}$ from scratch at every iteration by updating their components iteratively. With the exception of the first experiment 4.1, we further speedup our implementation by considering only the diagonal of the regularization term $\boldsymbol{H}^{(d)}$. All experiments were run on a Dell Inc. Latitude 7410 laptop with 16 GB of RAM and an Intel Core i7-10610U CPU running at 1.80 GHz. In all our experiments the Gaussian kernel $k\left(\boldsymbol{x}, \boldsymbol{x}'\right) = \exp\left(-||\boldsymbol{x}-\boldsymbol{x}'||_2^2/2l^2\right)$ was approximated by considering the Hilbert-GP [39] mapping of (6). In all experiments inputs were scaled to lie in a $D$-dimensional unit hypercube. When dealing with regression, the responses were standardized around the mean, while when considering

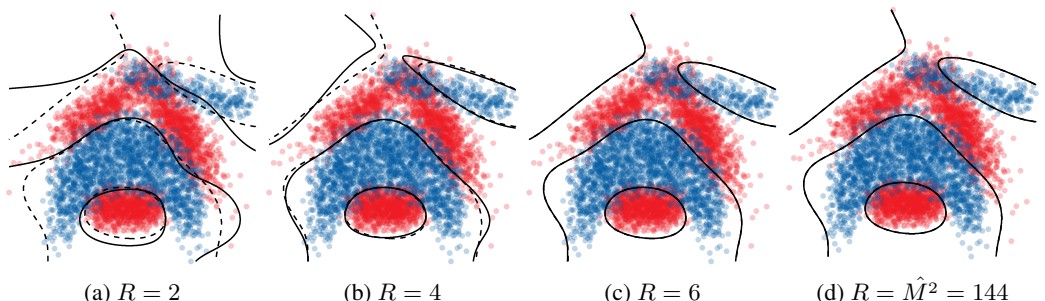

(a) $R = 2$    (b) $R = 4$    (c) $R = 6$    (d) $R = \hat{M}^2 = 144$

Figure 1: Classification boundary of the two-dimensional banana dataset for increasing CP-ranks. In the last plot the chosen CP-rank matches the true matrix rank. The dashed line is the Hilbert-GP, while the full line is T-KRR.

binary classification inference was done by looking at the sign of the model response (LS-SVM [41]). One sweep of our T-KRR algorithm is defined as updating factor matrices in the order $1 \to D$ and then back from $D \to 1$. All initial factor matrices were initialized with standard normal numbers and normalized by dividing all entries with their Frobenius norm. For all experiments the number of sweeps of T-KRR algorithm are set to 10. In the following three experiments it is shown how our proposed T-KRR algorithm is stable and recovers the full KRR estimate in case of low number of frequencies $\hat{M}$ and low rank $R$, consistently outperforming RFF. Finally, our model exhibits very competitive performance on large-scale problems when compared with other kernel methods.

## 4.1 Banana classification

The banana dataset is a two-dimensional classification problem [15] consisting of $N = 5300$ datapoints. Since the data is two-dimensional, we consider $\hat{M} = 12$ frequencies per dimension, which enables us to compare T-KRR with Hilbert-GP. Furthermore since for tensors of order two (i.e. matrices) the CP-rank is the matrix rank, we can deduce that our approach should recover the underlying Hilbert-GP method with $R = \hat{M}^2 = 144$. In this example we fix the kernel hyperparameters of our method and of the Hilbert-GP to $l = 1/2$ and $\lambda = 10^{-5}$ for visualization purposes. In Figure 1 we plot the decision boundary of T-KRR (full line) and of the associated Hilbert-GP (dashed line). We can see that already when $R = 6$ the learned decision boundary is indistinguishable to the one of Hilbert-GP, meaning that in this example it is possible to obtain a low-rank representation of the model weights. From a computational point of view, we solve a series of linear systems with $\hat{M}R$ unknowns instead of $\hat{M}^D$. However due to the low-dimensionality of the dataset, the computational savings per iteration are very modest, i.e. for $R = 6$ and $\hat{M} = 12$, we solve per iteration a linear system with 72 unknowns as opposed to the one-time solve of a linear system with 144 unknowns as in case of Hilbert-GP. As we show in Subsection 4.2, the linear computational complexity in $D$ of our algorithm results in ever-increasing performance benefits as the dimensionality of the problem becomes larger, where it becomes impossible to consider a full weight tensor.

## 4.2 Model performance with baseline

We consider five UCI [8] datasets in order to compare the performance of our model with RFF and the GPR/KRR baseline. For each dataset, we consider $90\%$ of the data for training and the remaining $10\%$ for testing. In particular, in all regression datasets, we first obtain an estimate of the lengthscale of the Gaussian kernel $l$ and regularization term $\lambda$ by log-marginal likelihood optimization over the whole training set using the GPLM toolbox [32]. We subsequently train GPR/KRR, RFF and our method with the estimated hyperparameter. In case of classification, we choose the lengthscale $l$ to be the sample mean of the sample standard deviation of our data (which is the default choice in the Matlab Machine Learning Toolbox and scikit-learn [30]) and $\lambda = 10^{-5}$. We repeat this procedure ten times over different random splits and report the sample mean and sample standard deviation of the predictive mean-squared error (MSE) and the misclassification rate for regression and classification respectively. In order to test the approximation capabilities of our approach, we set $\hat{M}$ and $R$ such

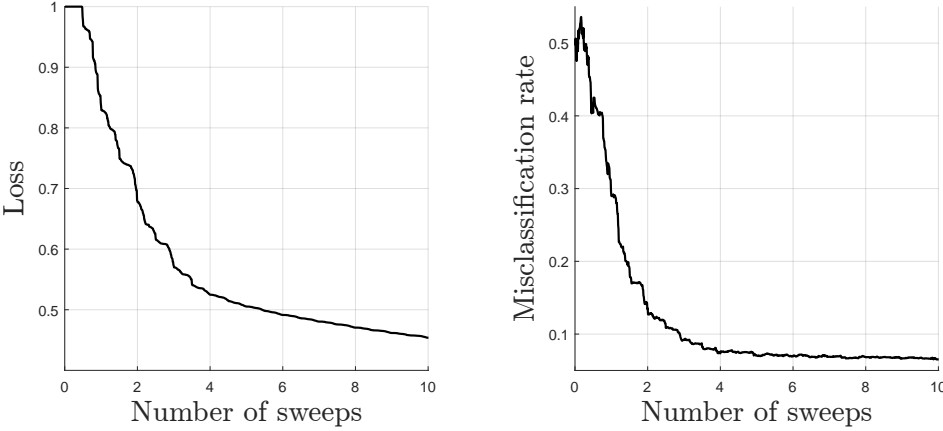

Figure 2: Normalized loss and misclassification rate while training on the Spambase dataset.

Table 1: Predictive MSE (regression) and misclassification rate (classification) with one standard deviation for RFF, T-KRR and KRR on different UCI datasets.

| Dataset | $N\downarrow$ | $D$ | $\hat{M}$ | $R$ | RFF | T-KRR | KRR |
|---------|------|-----|-----|-----|-----|-------|-----|
| Yacht | 308 | 6 | 10 | 25 | $0.0021 \pm 0.0018$ | $0.0009 \pm 0.0006$ | $\mathbf{0.0007} \pm 0.0004$ |
| Energy | 768 | 9 | 20 | 10 | $0.0275 \pm 0.0075$ | $0.0200 \pm 0.0066$ | $0.0200 \pm 0.0087$ |
| Airfoil | 1503 | 5 | 20 | 10 | $0.2180 \pm 0.0336$ | $0.1679 \pm 0.0258$ | $\mathbf{0.1587} \pm 0.0232$ |
| Spambase | 4601 | 57 | 40 | 10 | $0.3620 \pm 0.0188$ | $0.0935 \pm 0.0095$ | $\mathbf{0.0909} \pm 0.0115$ |
| Adult | 45222 | 96 | 40 | 10 | $0.3976 \pm 0.0056$ | $\mathbf{0.1596} \pm 0.0046$ | N/A |

that $\hat{M}R \ll N$ in order to benefit from computational gains. Furthermore, we note that due to the curse of dimensionality it is not possible to compare our approach with RFF while considering the same total number of basis functions $M = \hat{M}^D$. Hence we select $M_{\text{RFF}} = \hat{M}R$ such that the computational complexity of both methods is similar. Table 1 shows the MSE for different UCI datasets. We see that our approach comes close to the performance of full KRR and outperforms RFF on all datasets when considering the same number of model parameters. This is because our approach considers implicitly $\hat{M}^D$ frequencies to learn a low-rank parsimonious model which is fully described by $\hat{M}RD$ parameters, as opposed to RFF where the number of frequencies is the same as the number of model weights. Similarly to what is reported in [47, Figure 2] we observe lower performance of RFF on the Adult dataset than reported in [31]. Figure 2 plots the monotonically decreasing loss function as well as the corresponding misclassification rate while training with T-KRR on the Spambase dataset. After four sweeps the misclassification rate has converged. Since T-KRR is able to obtain similar performance as the KRR baseline on a range of small datasets, we proceed to tackle a large-scale regression problem.

## 4.3 Large-scale experiment

In order to highlight the ability of our method to deal with large data we consider the Airline dataset. The Airline dataset [14, 16] is a large-scale regression problem originally considered in [14] which is often used to compare state-of-the-art Gaussian Process Regression approximations due to its large size and its non-stationary features. The goal is in fact to predict the delay of a flight given eight features, which are the age of the airplane, route distance, airtime, departure time, arrival time, day of the week, day of the month, and month. We follow the same exact preprocessing steps as in the experiments in [16], [39] and [9], which consider subsets of data of size $N = 10000, 100000, 1000000, 5929413$, each chosen uniformly at random. Training the model is then accomplished with $2/3N$ datapoints, with the remaining portion reserved for testing. The entire procedure is then repeated 10 times with random initialization in order to obtain a sample mean and sample standard deviation estimate of the MSE. Following exactly the approach in Hilbert-GP [39], we consider $\hat{M} = 40$ basis functions per dimension, with the crucial difference that T-KRR

Table 2: Predictive MSE with one standard deviation for T-KRR. The prefix -A indicates that the model in question considers an additive kernel.

| $N$ | 10000 | 100000 | 1000000 | 5929413 |
|---|---|---|---|---|
| T-KRR ($R = 5$) | $0.91 \pm 0.10$ | $0.82 \pm 0.03$ | $0.80 \pm 0.02$ | $0.800 \pm 0.008$ |
| T-KRR ($R = 10$) | $0.89 \pm 0.05$ | $0.80 \pm 0.05$ | $0.79 \pm 0.02$ | $0.785 \pm 0.009$ |
| T-KRR ($R = 15$) | $0.90 \pm 0.07$ | $0.80 \pm 0.04$ | $0.78 \pm 0.02$ | $0.773 \pm 0.007$ |
| T-KRR ($R = 20$) | $0.97 \pm 0.15$ | $\mathbf{0.78} \pm 0.04$ | $\mathbf{0.77} \pm 0.01$ | $\mathbf{0.763} \pm 0.007$ |
| A-Hilbert-GP [39] | $0.97 \pm 0.14$ | $0.80 \pm 0.06$ | $0.83 \pm 0.02$ | $0.827 \pm 0.005$ |
| A-VFF [16] | $0.89 \pm 0.15$ | $0.82 \pm 0.05$ | $0.83 \pm 0.01$ | $0.827 \pm 0.004$ |
| SVIGP [14] | $0.89 \pm 0.16$ | $0.79 \pm 0.05$ | $0.79 \pm 0.01$ | $0.791 \pm 0.005$ |
| VISH [9] | $0.90 \pm 0.16$ | $0.81 \pm 0.05$ | $0.83 \pm 0.03$ | $0.834 \pm 0.055$ |
| GPR [33] | $0.89 \pm 0.16$ | N/A | N/A | N/A |
| A-GPR [10] | $0.89 \pm 0.16$ | N/A | N/A | N/A |

approximates a standard *product* Gaussian kernel, as opposed to an *additive* model. As was the case in the previous section for classification, we select the lengthscale of the kernel as the mean of the standard deviations of the eight features and choose $\lambda_N = {}^{100}/N$ for the different splits, where the dependency on $N$ is suggested by standard learning theory. We then train four distinct models for $R = 5, 10, 15, 20$. Table 2 compares T-KRR with the best performing models found in the literature, which all (except for GPR) rely on low-rank approximation of the kernel function. The T-KRR method is able to recover the baseline GPR performance already with $R = 5$ and remarkably outperform all other approaches although we choose the hyperparameters naively and consider equal lengthscales $l$ for all dimensions. Interestingly, a higher choice of $R$ does result in better performance in all cases except for $N = 10000$, where model performance very much depends on the random split (note that we consider different splits for each experiment). The SVIGP [14] achieves comparable results to T-KRR, indicating that a performance gain is possible from product kernels. The difficulty with SVIGP is however that the kernel function is interpolated *locally* at $M$ nodes in a data-dependent fashion, requiring an increasing amount of interpolation nodes to cover the whole domain to allow for good generalization. In contrast, T-KRR considers an exponentially large amount of basis functions in the frequency domain, and learns an efficient representation of the model weights. In light of the performance of the additive Hilbert-GP and additive VFF models, we expect similar performance when considering other feature maps which induce stationary kernels. When considering the whole dataset, training our model with $R = 5$ takes $1565 \pm 1$ seconds on a laptop, while for $R = 20$ it takes $7141 \pm 245$ seconds. Reported training times of SVIGP indicate $18360 \pm 360$ seconds [16] on a cluster. Since the computational complexity of our algorithm is dominated by matrix-matrix multiplications, we expect significant speedups when relying on GPU computations.

## 5 Conclusion

In this work a framework to perform large-scale supervised learning with tensor decompositions which leverages the tensor product structure of deterministic Fourier features was introduced. Concretely, a monotonically decreasing learning algorithm with linear complexity in both sample size and dimensionality was derived. This algorithm leverages the efficient format of the Canonical Polyadic Decomposition in combination with exponentially fast converging deterministic Fourier features which allow to implicitly approximate stationary product kernels up to machine precision. Numerical experiments show how the performance of the baseline Kernel Ridge Regression is recovered with a very limited number of parameters. The proposed method can handle problems which are both large in the number of samples as well as in their dimensionality, effectively enabling large-scale supervised learning with stationary product kernels. The biggest limitation of the current approach is that it is does not allow for uncertainty quantification, which motivates further work in that direction.

## Funding transparency statement

Frederiek Wesel, and thereby this work, is supported by the Delft University of Technology AI Labs program. The authors declare no competing interests.

## Acknowledgments

We would like to thank Arno Solin and Vincent Dutordoir for the help received with the Airline dataset.

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
