# A  Appendix

Derivation of the data-fitting term of (12). We make use of the multi-linearity property of the CPD and rely on re-ordering the summations:

$$\langle \mathcal{W}, \mathcal{Z}(\boldsymbol{x}) \rangle_{\mathrm{F}} = \left\langle \sum_{r=1}^{R} \boldsymbol{w}_r^{(1)} \otimes \boldsymbol{w}_r^{(2)} \otimes \cdots \otimes \boldsymbol{w}_r^{(D)}, \boldsymbol{z}^{(1)} \otimes \boldsymbol{z}^{(2)} \cdots \otimes \boldsymbol{z}^{(D)} \right\rangle_{\mathrm{F}}$$

$$= \sum_{i_1=1}^{\hat{M}} \cdots \sum_{i_d=1}^{\hat{M}} \cdots \sum_{i_D=1}^{\hat{M}} \sum_{r=1}^{R} w_{i_1 r}^{(1)} z_{i_1}^{(1)} \cdots w_{i_d r}^{(d)} z_{i_d}^{(d)} \cdots w_{i_D r}^{(D)} z_{i_D}^{(D)}$$

$$= \sum_{i_d=1}^{\hat{M}} \sum_{r=1}^{R} w_{i_d r}^{(d)} \left( z_{i_d}^{(d)} \sum_{i_1=1}^{\hat{M}} w_{i_1 r}^{(1)} z_{i_1}^{(1)} \cdots \sum_{i_D=1}^{\hat{M}} w_{i_D r}^{(D)} z_{i_D}^{(D)} \right)$$

$$= \mathrm{vec}\left( \boldsymbol{W}^{(d)} \right)^{\mathrm{T}} \left( \boldsymbol{z}^{(d)} \otimes \left( \boldsymbol{z}^{(1)^{\mathrm{T}}} \boldsymbol{W}^{(1)^{\mathrm{T}}} \odot \cdots \odot \boldsymbol{z}^{(D)^{\mathrm{T}}} \boldsymbol{W}^{(D)^{\mathrm{T}}} \right) \right)$$

$$= \left\langle \mathrm{vec}\left( \boldsymbol{W}^{(d)} \right), \boldsymbol{g}^{(d)}(\boldsymbol{x}) \right\rangle$$

The derivation of the regularization term of (13) follows a similar reasoning as for the data-fitting term:

$$\langle \mathcal{W}, \mathcal{W} \rangle_{\mathrm{F}} = \left\langle \sum_{r=1}^{R} \boldsymbol{w}_r^{(1)} \otimes \boldsymbol{w}_r^{(2)} \otimes \cdots \otimes \boldsymbol{w}_r^{(D)}, \sum_{r=1}^{R} \boldsymbol{w}_r^{(1)} \otimes \boldsymbol{w}_r^{(2)} \otimes \cdots \otimes \boldsymbol{w}_r^{(D)} \right\rangle_{\mathrm{F}}$$

$$= \sum_{i_1=1}^{\hat{M}} \cdots \sum_{i_d=1}^{\hat{M}} \cdots \sum_{i_D=1}^{\hat{M}} \sum_{r=1}^{R} \sum_{p=1}^{R} w_{i_1 r}^{(1)} w_{i_1,p}^{(1)} \cdots w_{i_d r}^{(d)} w_{i_d,p}^{(d)} \cdots w_{i_D r}^{(D)} w_{p i_D}^{(D)}$$

$$= \sum_{r=1}^{R} \sum_{p=1}^{R} \left( \sum_{i_d=1}^{\hat{M}} w_{i_d r}^{(d)} w_{i_d p}^{(d)} \right) \left( \sum_{i_1=1}^{\hat{M}} w_{i_1 r}^{(1)} w_{i_1 p}^{(1)} \cdots \sum_{i_D=1}^{\hat{M}} w_{i_D r}^{(D)} w_{i_D p}^{(D)} \right)$$

$$= \sum_{r=1}^{R} \sum_{p=1}^{R} \left( \boldsymbol{w}_r^{(d)^{\mathrm{T}}} \boldsymbol{w}_p^{(d)} \right) \left( \boldsymbol{w}_r^{(1)^{\mathrm{T}}} \boldsymbol{w}_p^{(1)} \odot \cdots \odot \boldsymbol{w}_r^{(D)^{\mathrm{T}}} \boldsymbol{w}_p^{(D)} \right)$$

$$= \mathrm{vec}\left( \boldsymbol{W}^{(d)^{\mathrm{T}}} \boldsymbol{W}^{(d)} \right)^{\mathrm{T}} \mathrm{vec}\left( \boldsymbol{W}^{(1)^{\mathrm{T}}} \boldsymbol{W}^{(1)} \odot \cdots \odot \boldsymbol{W}^{(D)^{\mathrm{T}}} \boldsymbol{W}^{(D)} \right)$$

$$= \left\langle \mathrm{vec}\left( \boldsymbol{W}^{(d)^{\mathrm{T}}} \boldsymbol{W}^{(d)} \right), \mathrm{vec}\left( \boldsymbol{H}^{(d)} \right) \right\rangle$$