# OpenReview forum: "Large-Scale Learning with Fourier Features and Tensor Decompositions"
_NeurIPS.cc/2021/Conference — NeurIPS 2021 Poster_

### Official Review · Reviewer_LNEF · 2021-07-14

**Rating:** 6
**Confidence:** 4

**Summary:**

This work proposes to use tensor decomposition techniques to approximate the deterministic Fourier features (DFF), in large-scale kernel learning. The main idea is that, after the DFFs are generated in the tensor form, low-rank approximation by tensor decomposition is applied, which can be formulated as a regularized convex optimization. Block coordinate descent is then used to solve the problem effectively. Experiments on kernel ridge regression and Gaussian process regression with synthetic and real datasets confirm the effeciency of the proposed approach.

**Main Review:**

The paper is written very clearly, and the organization is neat, so it's very easy to follow.

In general, the problem studied in the work, speeding up large-scale kernel learning, is valuable and interesting to many people in the NeurIPS community. Deterministic random feature have received less attention before since it is harder to be applied in practice and suffers from the curse of dimensionality, but as the paper shows, it has its own advantages and can be competible in certain problems. Tensor dedcomposition is also a trending topic recently. The idea of combining them together is a good attempt.

Regarding the content of the paper, the first 5 pages are rather fundamental and mostly introduce existing concepts and algorithms.  The problem formulation is presented in Section 3.4, which is also quite standard and does not require novel techniques. Hence, I would say that this paper is more like an experimental/application paper, that applies existing methods in tensor analysis to deterministic Fourier features.

Questions:

1) Could please specify the dimensionality of the feature space of proposed low-rank DFF? Should it be $\hat MRD$ or $\hat MR$ (line 207 and 278)? This relates to the validity of comparison with RFF.

2) How does RFF compare with T-KRR in terms of exact running time? What is the optimization algorithm for RFF?

3) What is the relationship between Hilbert-GP, DFF and the proposed method? There are some description in e.g. sec 4.1, but I suggest to make the comparison and relation with other methods more clear, e.g., list the complexities and equivalence conditions explicitly.

4) Besides computational complexity, storage efficiency might also be an important topic. I believe the low-rank approximation is also beneficial in terms of memory in some way. On this regard, there are some works on memory-efficient RFF that might be related,

Data-dependent compression of random features for large-scale kernel approximation, AISTATS 2019

Low-Precision Random Fourier Features for Memory-constrained Kernel Approximation, AISTATS 2019

Quantization Algorithms for Random Fourier Features, ICML 2021

In all, I think this paper is a good attempt to apply tensor analysis to machine learning to solve practical problems. If the authors can answer my questions with satisfactory, I would recommend a weak acceptance.

**Time Spent Reviewing:**

2

---

> ### Author Response · Authors · 2021-08-06
> **Answer to Official Review of Paper3141 by Reviewer LNEF**
>
> We thank the reviewer for his or her useful remark and suggestions, some of which we would like to address:
>
> 1) Regarding the content of the paper, the first 5 pages are rather fundamental and mostly introduce existing concepts and algorithms.
>
> In our opinion, the different tensor concepts and the explicit form of the tensor product approximation of the Gaussian kernel (published in 2020) were explained in this article as these concepts are not widely known, especially not in a broad conference such as NeurIPS.
>
> 2) The problem formulation is presented in Section 3.4, which is also quite standard and does not require novel techniques.
>
> We respectfully disagree with this statement. Section 3.4 is in our opinion not standard. The problem formulation given in equations (10) and (11) is exactly our main contribution. We exploit, for the first time to our knowledge, the tensor product structure of the feature map to efficiently solve the primal problem.
>
> 3) Could please specify the dimensionality of the feature space of proposed low-rank DFF? Should it be $\hat{M}RD$ or $\hat{M}R$? This relates to the validity of comparison with RFF.
>
> The dimensionality of the considered DFF feature space is $\hat{M}^D$. In the comparison with RFF, we draw $\hat{M}R$ frequencies so that the computational complexity of one iteration of our method is the same as RFF. Picking $\hat{M}RD$ frequencies does not relevantly improve the RFF results, but if the reviewers think this invalidates the comparison, we would be happy to rerun updated experiments.
>
> 4) How does RFF compare with T-KRR in terms of exact running time? What is the optimization algorithm for RFF?
>
> The running times of RFF are $D$ times faster than T-KRR, as one can deduce from the complexity analysis ($D$ linear systems are set-up and solved instead of one), however RFF considers a feature space of dimensionality $\hat{M}$ instead of $\hat{M}^D$ for T-KRR. Therefore we decided not to include the running times in Section 4.2. The linear system obtained by the matrix inversion lemma related to the RFF approximation of the full problem is solved numerically with a direct solver\~(MATLAB backslash).\\
>
> 5) What is the relationship between Hilbert-GP, DFF and the proposed method? There are some description in e.g. sec 4.1, but I suggest to make the comparison and relation with other methods more clear, e.g., list the complexities and equivalence conditions explicitly.
>
> We thank the reviewer for this valuable suggestion. DFF recovers the full posterior mean GP estimator (same as KRR, same as LS-SVM) in the limit of the number of frequencies $M$. Our approach recovers the DFF estimator when $R$ is equal to the true rank of the weight tensor (there are many upper bounds in literature regarding $R$). In Section 4.1, since the weight tensor collapses to a matrix, we do know $R$ exactly and we can illustrate how a low-rank approximation of the weight tensor leads to equally good results. This comparison will be added to a future version of the article.
>
>
> 6) Besides computational complexity, storage efficiency might also be an important topic. I believe the low-rank approximation is also beneficial in terms of memory in some way. On this regard, there are some works on memory-efficient RFF that might be related
>
> We agree with the reviewer and would like to thank him or her for the suggestion. In a naive implementation, the memory costs are dominated by the construction of the matrix containing the mapped inputs $G^{(d)}$, which has size $N\times \hat{M}R$. In practice however, just like any low rank approximation, when making $(G^{(d)})^T G^{(d)}$ one can compute this in chunks, limiting the necessary memory to $\left(\hat{M} R\right)^2$ blocks. In our approach, it is also not necessary to store all $G^{(d)}$, but one copy can conveniently be updated each iteration. It is necessary to store at all times the weight tensor in decomposed form, which costs $\hat{M}RD$ memory blocks. Storage complexity of our method will be discussed in more detail in an updated version of this article. We thank the reviewer for the references. Limiting the precision of the computations could also be applied to our method, resulting in further storage complexity reduction. Also, given the large quantity of stochastic feature maps, we plan a more detailed comparison with existing stochastic approximations.

---

> > ### Comment · Reviewer_LNEF · 2021-08-26
> > **Thanks for the response**
> >
> > Thanks for the response. I will keep my score. I have a few suggestions when you revise the paper:
> >
> > 1. Compare the dimensionality and complexity of the proposed method with others (e.g. full-rank, RFF, Nystrom etc) more clearly, perhaps in a table.
> > 2. Discuss the relationship of the proposed method with Hilbert-GP, DFF etc in the related work section.
> > 3. Add some discussion on the memory complexity and efficiency.
> >
> > Also, as the authors claimed, a more detailed numerical comparison with more stochastic approximation methods would surely be better.

---

### Official Review · Reviewer_kD5f · 2021-07-14

**Rating:** 6
**Confidence:** 3

**Summary:**

In this paper, the authors propose a novel kernel approximation scheme based on deterministic Fourier features and tensor decomposition. The proposed approach improves previous efforts in the sense that (i) it deduces the (kernel) approximation error to decrease exponentially with the number of frequencies (or Fourier features); and (ii) it overcomes the ”curse of dimensionality“ with the low-rank tensor decomposition approach, the complexity of which scales linearly in the input dimension. Simulation results (on the performance of tensor kernel ridge regression) are provided to validate the proposed method, on both small and large dimensional datasets.

**Limitations And Societal Impact:**

The authors have adequately addressed the possible limitations of their approach. This work is mainly theoretical and algorithmic and I do not see any possible negative societal impact.


**Main Review:**

The article is in general well written and makes a solid contribution to the approximation of large-scale kernel matrices. A few comments:
* Only performance of kernel-based methods are compared on various datasets, e.g., MSE in the regression setting or misclassification rate in the classification setting. Unless I missed something, the proposed method is designed to (i) deduce the **approximation error** (with respect to the underlying kernel matrix) in a more efficient manner and (ii) reduce the (storage and computational) **complexity**. None of the numerical results seems to reflect this information/improvement.
* It seems that the defining equation of G^(d) should be G^d?


================================================
After rebuttal: I thank the authors for their clarification, that helps. I believe that adding more experiments on the gain of (storage and computational) **complexity** with the proposed method will broaden the impact of this work (for the moment there are only two sentences on the time complexity). My score remains the same.

**Time Spent Reviewing:**

2

---

> ### Author Response · Authors · 2021-08-06
> **Answer to Official Review of Paper3141 by Reviewer kD5f**
>
> We thank the reviewer for his or her useful remarks, which we would like to address.
>
> 1) Unless I missed something, the proposed method is designed to deduce the approximation error (with respect to the underlying kernel matrix) in a more efficient manner
>
> The main contribution is not to reduce the approximation error (with respect to the underlying kernel matrix) in a more efficient manner. The proposed contribution lies in solving the primal problem efficiently by exploiting the tensor product structure of the feature map (which does indeed converge efficiently). However, in no case do we ever explicitly construct a kernel matrix. Therefore, no approximation error with respect to the underlying kernel matrix is reported in the numerical experiments.
>
> 2) [...] storage and computational complexity. None of the numerical results seems to reflect this information/improvement.
> In Section 4.3 we compared the training time of our method to the state of the art. ``When considering the whole dataset, training our model with $R=5$ takes $1565\pm1$ seconds on a laptop, while for $R=20$ it takes $7141\pm245$ seconds. Reported training times of SVIGP indicate $18360\pm360$ seconds on a cluster.''
> With regards to storage complexity, we think this is a very valuable point and we will definitely report this in an updated version of our article.
>
> 3) It seems that the defining equation of $G^{(d)}$ should be $G^d$?
> We thank the reviewer for noting the typo.

---

### Official Review · Reviewer_pffh · 2021-07-16

**Rating:** 8
**Confidence:** 4

**Summary:**

The authors propose a method for inference of large-scale gaussian process (GP) regressio problems with relatively high-dimensional inputs. The method achieves state of the art performance on real-world datasets using 2 innovations 1) formulation of the regression problem in the primal form with deterministic Fourier features, 2) reducing the dimensionality of the regression weights in the primal space by  defining them as a low-rank tensor. The authors describe the advantages and disadvantages of different choices for the tensor decomposition and demonstrate computational and inferential performance on both synthetic and real datasets. Of note is their ability to achieve comparable performance using a laptop computer to competing methods using clusters.

**Limitations And Societal Impact:**

Limitations were addressed but there was no mention of societal impacts.

**Main Review:**

The paper is well-motivated, accessibly written, and well demonstrated. I have very little critical to say about it other than that I'm not sure how this approach helps with hyperparameter learning, which tends to be some of the highest computational overhead of GP methods. Did I miss something?

Finally, the authors mention the non-probabilistic nature of these methods. They might think of their method in the context of work by Rose Yu and colleagues, starting with their 2018 AISTATS paper. Although they rely on the Tucker decomposition, rather than CP decomposition, it might be closely-related enough to consider mentioning.


**Time Spent Reviewing:**

2

---

> ### Author Response · Authors · 2021-08-06
> **Answer to Official Review of Paper3141 by Reviewer pffh**
>
> We thank the reviewer for his or her very useful remarks. Indeed our approach, since it is non-probabilistic, does not allow for GP-like hyperparameter optimization. Hyperparameter selection can be carried out via cross-validation as learning is cheap. We should point this out in a future version of this article. We plan to do further work in that direction by developing a probabilistic model which allows for joint hyperparameter learning. We thank the reviewer for the very interesting Rose Yu paper which we had missed.

---

### Official Review · Reviewer_h3fL · 2021-07-19

**Rating:** 6
**Confidence:** 3

**Summary:**

Authors present a tensor kernel ridge regression method where the covariance kernel is replaced with random Fourier features to reduce the typical O(N^3) cost down to O(NM^2) for M random frequencies. RFFs typically have slow convergence in approximating the full kernel w.r.t. to the number of random frequencies. The tensor is assumed to have a rank-R CPD decomposition to further reduce the complexity of the problem.

**Limitations And Societal Impact:**

The authors have adequately addressed their limitations. I do not see any potential negative societal impacts.

**Main Review:**

I think the fundamental idea is solid but I have the following questions about the implementation of the model

1.) In Section 4.2 you mention that you use a GP to learn the hyperparameters of your model, wouldn't this contradict the main point of your paper? Why not include the hyperparameter learning component as a part of the learning procedure for your model and the comparisons?
2.) In Section 4.3, why not actually learn hyperparameters in some manner that is non-naive? There is little to suggest that this strategy generalizes to a broad scenario. I think you need to discuss in detail how to learn the hyperparameters for your approach



**Time Spent Reviewing:**

5

---

> ### Author Response · Authors · 2021-08-06
> **Answer to NeurIPS 2021 Conference Paper3141 Reviewer h3fL**
>
> We thank the reviewer for his or her very useful remarks. We would like at first to point out that our approach does not rely on random Fourier Features. In fact, we consider Deterministic Fourier Features, which converge to the Kernel function in question exponentially fast, in contrast with Random Fourier Features (which converge at the slow Monte Carlo rate). To address the main problem of Deterministic Fourier Features, which is the exponential grow of the number of model parameters as a function of the dimensionality of the problem, we model the model parameters as a low-rank tensor. This allows us to consider an exponentially large $M=\hat{M}^D$ number of basis functions (which achieve fast convergence) while having linear training time in both sample size and dimensionality. Training time is thus of $O(N \hat{M}^2R^2)$, instead of $O(N (\hat{M}^D)^2)$ of Random Fourier Features.
>
> We address the remarks:
> 1) In Section 4.2 you mention that you use a GP to learn the hyperparameters of your model, wouldn't this contradict the main point of your paper? Why not include the hyperparameter learning component as a part of the learning procedure for your model and the comparisons?
>
> In Section 4.2 we indeed compare our method with a GP in case of regression. We fix the hyperparameters of our method (which in this case are just the lengthscale of the Gaussian kernel and the regularization term) to match the ones of the GP. We do this in order to investigate whether and to which extent the performance of our approach recovers the one of the full GP fixing the hyperparameters. We expect this to happen in theory, since with enough basis functions the kernel function is approximated almost to machine precision. The remaining question is then if a CP rank-R approximation of the model parameters is able to provide a good enough representation of the model parameters. Our experiments set-up in this way show that it is indeed the case.
>
> 2) In Section 4.3, why not actually learn hyperparameters in some manner that is non-naive? There is little to suggest that this strategy generalizes to a broad scenario. I think you need to discuss in detail how to learn the hyperparameters for your approach
>
> As in case of classical Kernel Ridge Regression, hyperparameters can be chosen if one wants via cross-validation. In Section 4.3 we opted due to time pressure indeed for a very standard and uninformative choice of the lengthscale to demonstrate the efficacy of our method. This approach was followed in case of classification in Section 4.2 with satisfactory results. We believe that cross-validation would further improve the performance, and we agree with the reviewer: this point would be discussed in a future version of this article.

---

### Decision · Program_Chairs · 2021-09-27

**Decision:**

Accept (Poster)

**Comment:**

This paper gives a deterministic version of the random feature model by constructing the features explicitly using certain tensor structure. The additional structure allows more efficient computation and performs well in practice. The reviewers found that the experiments are convincing and the ideas are sound. There were some confusions about the exact complexity of the proposed algorithm, which should be clarified in the revised paper.